# Optimal Design of Plated/Shell Structures under Flutter Constraints—A Literature Review

**DOI:** 10.3390/ma12244215

**Published:** 2019-12-15

**Authors:** Aleksander Muc, Justyna Flis, Marcin Augustyn

**Affiliations:** Institute of Machine Design, Cracow University of Technology, 31-864 Cracow, Poland; olekmuc@mech.pk.edu.pl (A.M.); augustyn@mech.pk.edu.pl (M.A.)

**Keywords:** optimal design, flutter, aeroelastic behavior, plates, shells, advanced materials

## Abstract

Aeroelastic optimization has become an indispensable component in the evaluation of divergence and flutter characteristics for plated/shell structures. The present paper intends to review the fundamental trends and dominant approaches in the optimal design of engineering constructions. A special attention is focused on the formulation of objective functions/functional and the definition of physical (material) variables, particularly in view of composite materials understood in the broader sense as not only multilayered laminates but also as sandwich structures, nanocomposites, functionally graded materials, and materials with piezoelectric actuators/sensors. Moreover, various original aspects of optimization problems of composite structures are demonstrated, discussed, and reviewed in depth.


**Contents**
**1.** 
**Introduction**
**2.** 
**Brief Description of the Flutter Problem**
**3.** 
**Remarks on the Formulation of Optimization Problems**
**4.** 
**Objective Functions**
*4.1.* 
*Deterministic Approach*
*4.2.* 
*Reliability Analysis*
**5.** 
**Physical (Material) Design Variables**
*5.1.* 
*Composite Materials*
*5.2.* 
*Functionally Graded Materials and Nanocomposite Structures—Thermal Protection*
*5.3.* 
*Piezoelectric (PZT) Patches–Active and Passive Flutter Control*
*5.4.* 
*Sandwich Structures*
**6.** 
**Geometric Design Variables**
*6.1.* 
*Cross-Section Parameters–Variable (Stepped) Thickness*
*6.2.* 
*Form of the Structure*
**7.** 
**Numerical (Finite Element) Packages**
**8.** 
**Optimization Algorithms**
**9.** 
**Concluding Remarks**



## 1. Introduction

The concept of the flutter can be described as an instability associated with self-sustained and self-excited vibration which is a combination of elastic, inertial, and aerodynamic forces where the structure and the flow around it interact with each other. The terms “self-sustained” and “self-excited” mean respectively “without external constraint” and “without unsteadiness coming from upstream or downstream”. Flutter results in energy exchange between the structure and the fluid, when the structure is self-excited, its vibration induces an unsteady pressure field around the profile sustaining the vibration. The flutter problem usually starts from small mechanical or aerodynamic disturbance above a critical flow velocity, then gives large vibration amplitudes and finally damages in a short period of time.

This phenomenon is a significant problem encountered in the design of aircraft structures [1,2] or turbine blades in the turbomachines (Srinivasan [3]). Flutter is thus a major concern for the designers regarding both the safety and costs.

The current aerospace industry demands innovative designs and materials that offer weight savings, as well as faster and more cost and energy efficient structures. To fulfill these needs from the structural point of view, researchers have proposed the use of laminated composite materials, composite materials with curvilinear fibers, functionally graded materials (FGM), sandwich structures, stiffeners, actuators etc. Aeroelasticity is a crucial discipline in the design and maintenance of an aircraft. At high speeds, the wing twists and vibrates up and down changing the angle of attack. The change in the angle of attack induces a change of the aerodynamic forces, which subsequently affects the angle of attack, resulting in a continuously vibrating system. The phenomenon of self-excited oscillation of a structure, which extracts energy from the airstream, is called aeroelastic flutter. Currently, although numerical tools are more and more reliable, flutter prediction still depends on simplified models and systematic tests. 

In the flutter analysis the attention is mainly focused on the discussion of different problems that can affect the structural behavior, i.e., the aerodynamic theories, the form of boundary conditions, the structural geometry (the analysis deals mainly with 2D structures), the material properties and the effects of aerothermoelastic coupling.

Optimal design of flutter characteristics of plated/shell structures is a significant and interesting engineering problem which, in our opinion, is rather poorly investigated in the last decades. For instance, in the Sonmez’s review [4] on optimum design of composite structures presented only 5/1007 (flutter speed) and 7/1007 (aerodynamic performance) works. Alijani, Amabili [5] discussed 6/314 works dealing with flutter problems.

The review is generally organized as follows: the brief introduction to the flutter problems is presented in Section 2. In Section 3 the general formulation of the optimization problems is given in order to emphasize the fundamental elements playing the most important role in the optimal design, i.e., the objective functions/functional (Section 4) and design variables—physical/material (Section 5) and geometrical (Section 6). Section 7 is devoted to the short presentation of the existing commercial finite element packages that may be used in the evaluation of the objective functions. The above information is supplemented by the discussion of optimization algorithms applied in the analysis—Section 8. Concluding remarks describing especially the present state-of-art in the area of optimization plated/shell constructions under flutter constraints are presented in Section 9. 

The fundamental aim of the present paper is to review the existing literature on optimization of constructions subjected to flutter constraints. A special attention is focused on the proper choice of material properties since they have an important effect on flutter behavior of turbine blades or aircraft wings.

## 2. Brief Description of the Flutter Problem

Flutter problem is a complex aerodynamic phenomenon that deals not only with the description of the dynamic behavior of structures but especially with the characterization of the fluid-structure interaction arising during the flow of a fluid over a surface of a construction considered. Table 1 summarizes the possible classification of models used in the analysis.

Using the linear piston theory, the aerodynamic pressure acting on a curved surface area dA is given by:(1)$$\begin{array}{l} \Delta p=-\Lambda \left(\frac{\partial w}{\partial x}\cos\vartheta_{\infty }+\frac{\partial w}{\partial y}\sin\vartheta_{\infty }\right)-\mu \frac{\partial w}{\partial t},\\ \Lambda =\rho_{\infty }V_{\infty }^{2}/\sqrt{M_{}^{2}-1},\;\mu =\rho_{\infty }V_{\infty }\left(M_{}^{2}-2\right)/{\left(M_{}^{2}-1\right)}^{3/2} \end{array}$$
in which $\rho_{\infty },\;V_{\infty }$, and *M* are the free stream air density, velocity of air, Mach number, respectively, and $\vartheta_{\infty }$ denotes the flow angle. *x* and *y* denote the curvilinear coordinates. Forsching [6] introduced available conditions of using piston theories.

To perform the stability analysis that takes into account the aerodynamic interactions (1) the fundamental set of equations can be reduced to the following form:(2)$$L_{1}w\left(x,y,t\right)=L_{2}\left(\Delta p+\rho h\frac{\partial^{2}w}{\partial t^{2}}\right)$$
*L*_1_, *L*_2_ are linear differential operators with respect to *x*, *y* variables, *w* denotes the normal to the mid-surface displacements, *t* is a time and *h* means the thickness. For the Love-Kirchhoff kinematical hypothesis the explicit form of the differential operators is presented by Bolotin [7,8], Hedgepeth [9], Houbolt [10], Sawyer [11], Bohon [12], Stepanov [13], Muc, Flis [14].

Introducing the Airy stress functions Φ(*x*,*y*,*t*) the relation (2) can be written in the equivalent form in terms of the variable Φ(*x*,*y*,*t*)–see Bolotin [7,8], Rikards, Teters [15].

The use of the first-order transverse shear deformation theory changes the form of differential operators *L*_1_, *L*_2_ but not their order—see e.g., Li, Song [16]. 

It is worth to note that the relation (2) describes also the thermoelastic and piezoelectric effects by the introduction of the appropriate terms to the functional of the external forces (mechanical, thermal, and electric forces)—see Muc, Flis [14], and Muc et al. [17].

Usually, the flutter phenomenon is characterized by the plots of variations of natural frequencies with aerodynamic pressures—Figure 1. The neighborhood modes approach to each other at the value of the critical aerodynamic pressure that corresponds to the loss of stability. The values of the critical aerodynamic pressures are different for various vibrational modes. As it is shown in Figure 1 the mechanical (material) properties may change drastically both the pre-flutter and post-flutter behavior—the modes of frequency coalescence may be totally different for isotropic and anisotropic structures.

The fluttering mode is completely different that natural vibration modes of the structure—see the graphical representation shown in Figure 2.

The flutter characteristics, demonstrated schematically in Figure 1 and Figure 2, are functions of many parameters. Their effects will be discussed later in view of the optimal design. Early reviews, dealing particularly with parametric investigations, on plate/shell flutter are presented by Fung [18], Dowell [19]. Recent, more comprehensive review of researches in the area of flutter behavior of structures can be found in Pettit [20] and Kouchakzadeh et al. [21].

## 3. Remarks on the Formulation of Optimization Problems

Optimization is increasingly entering many practical engineering tasks, which may cause problems related to the use of appropriate terms and understanding of many concepts. Because of the above-mentioned facts, we will introduce a separate classification of optimization tasks solved later in this work. It mainly covers the problems of structural optimization.

The detailed information about the formulation of various optimization problems can be found in Reference [22].

To formulate an optimization problem it is necessary to define the following quantities/parameters/functions/functionals:A vector of design variables and a space of design variables;An objective function or an objective functional;A set of constraints in the form of equality or inequality.

In this section, we intend to present only general aspects of this problem, especially those related to the numerical procedures used by the authors. Detailed formulations of specific optimization problems are given in the next sections.

The solution of any engineering problem is determined by a set of independent variables, which can be called design variables $\vec{s}$ not only in optimization problems. We assume that the vector $\vec{s}$ has I independent components. Depending on the formulation of the optimization problem, each of the design variables *s_i_* can be represented as:

Continuous real variable, i.e.,
*s_i_* ∈ R, *i* = 1, 2, …, *I*(3)

Discrete variable, i.e.,
*s_i_* ∈ C = {c_1_,c_2_…}, *i* = 1, 2, …, *I*; c_p_ ∈ R or c_p_ ∈ N(4)

The spaces of decision variables defined by formulas (3) and (4) (the superscript means the type of design variable to be used) are able to describe all possible changes in the design variables considered in a given optimization problem and are therefore referred to free search space (FSS), in contrast to the area on which constraints are imposed.

Constraints (in the form of inequality or equality) resulting from the analysis of the physical phenomenon or due to technological considerations are the only limitations of the optimization problems analyzed herein. They can be written the following way:
(5)$$g_{i}(\vec{s})\;\leq \;0,\;j=1,2,\;\ldots \;J$$
(6)$$h_{k}(\vec{s})\;\leq \;0,\;k\;=\;1,2,\;\ldots \;K$$

Again, for generality of the numerical procedures developed herein, we formulate the optimization problem as searching for the maximum of the objective function/functional *f*:(7)$$Max\;f\left(\vec{s}\right)$$

In order to generalize the numerical analysis of various optimization problems discussed in the further part of the work, the *Min* or *MinMax* search problem is transformed in a classical way (only in a numerical program) to the task of searching for the maximum.

For a given type of space of design variables S ((3) or (4)), the optimization problem (5)–(7) will be expressed through three symbols:(8)$$\langle S,f,\phi \rangle \;\mathrm{where}\;\phi \left(s\right)=\{\begin{array}{l} 1\;(\mathrm{true})\;\mathrm{conditions}\;(5),\;(6)\;\mathrm{are}\;\mathrm{satisfied}\\ 0\;(\mathrm{false})\;\mathrm{conditions}\;(5),\;(6)\;\mathrm{are}\;\mathrm{not}\;\mathrm{satisfied} \end{array}$$
where *ϕ*(*s*) is a Boolean function and S denotes the space of design variables.

A different classification method is associated with the type of design variables defined as both real or discrete variables. Generally, we can distinguish three types of variables:In dimensional (parametric) optimization, design variables determine the structure thickness distribution and its parameters characterizing the cross-section;In shape optimization, these are the describing variables:
i.The geometry (and thus also the shape) of the outer edge of the structure;ii.The geometry of the mid-surface of the structures (beams, plates, shells).iii.In the optimization of the topological structure, the design variables define:iv.The manner of connection of elements, areas, or components of the structure;v.The number and spatial distribution of structure elements;vi.Material distribution in the structure.

In designing the optimal topology of the structure, the set of design variables may change in the optimization process.

The above definition was introduced by Kirsch [23] in 1989 and identical classification is presented in monograph [24] published in 2002. The author of the cited works indicates that in many cases the problems are being solved, in which changes of topological, geometric, and dimensional variables are simultaneously made (see e.g., [25]) and such tasks are called lay-out optimization in English, which we think it is best to translate it as structure optimization. However, at the end of the 1980 s, Bendsøe [26,27] introduced the concept of topological optimization, which means also the changes in the material distribution in the structure. In monograph [28], it is explained that in fact topological optimization is understood as the optimization of the material structure constituting the structure, in the sense of searching for the optimal material density ρ(x) associated with the mechanical properties of the material through the relation:(9)$$C_{ijkl}(x)=\rho (x)C_{ijkl}^{0}$$
with a constraint condition in the form of:(10)$$\int_{\Omega }\rho (x)\leq V$$
where [C] is the stiffness matrix. The concept of topological optimization therefore includes a full set of design variables that combine dimension, shape, topological, and material-related design variables. However, this above generalization should not be confused with a strictly understood definition of topology and understood strictly in the theory of optimization of topological variables.

In the optimization of constructions made of composite materials, lay-out optimization of composite structures is the most adequate instead of topological optimization. This is due to the fact that the topological optimization is mainly associated with the term of the material homogenization. In our opinion, the most reasonable seems to be the use of the definition: the optimization of the structure of composite materials with the simultaneous distinction of the level of analysis, i.e., it is possible to optimize at the level:Elementary cell,Individual layer,Laminate.

This is necessary because on each of the above-mentioned levels of analysis, CM introduce different types of design variables including both the geometry and the type of reinforcement, matrix, and interfacial layer or the orientation of individual layers, individual thickness of individual layers, and their material properties. The set of design variables describing the structure of CM is an arrangement of geometrical and physical variables that characterize (at a given level of analysis) unambiguously the physical properties of CM. The next difference between the definitions proposed in this book and the Bendsøe approach [27] lies in the possibility of describing the structure by other design variables than those that characterize the problem’s physics, namely dimensional and shape optimization variables describing the geometry of the problem. In this sense, we use two types of design variables at the present work:Physical (material) representing the CM structure from which the structure is made;Geometric—characterizing the geometry of the structure.

The concept of physical variables relates directly to the issues of material design. It refers to the design of physical properties of a composite material by changing the matrix and reinforcement material as well as the geometrical features of the reinforcement. In the second case, this can be done by changing the size, shape, or distribution of the reinforcing elements or only their orientation and stacking sequences.

In the case of geometric variables, we can proceed in two ways:Only by selecting the material when looking for the optimal distribution of the laminate thickness, the thickness and shape of the reinforcing patches, the shape of the middle surface of the structure, or the shape of the edge;By designing a new material, if in the above problems there are additional constraints (technological, geometric, etc.,) that none of the currently available materials meet.

It should be emphasized that the work also uses the term laminate configuration regarding a set of design variables describing the CM structure at the lamination level and referring directly to the term laminate configuration used in the Anglo-Saxon literature. This concept should not be confused with the term combinatorial configuration introduced in mathematical combinatorial, because in mathematics it has a completely different meaning.

## 4. Objective Functions

Basic mathematical formulations of design objectives are subjected to various uncertainties inherent in structural, material, damage parameters, etc., which are modeled by random variables. They can be taken into account (or not), so that possible definitions of design objectives are divided into two groups, i.e., deterministic and reliability based optimization problems. In literature both formulations are treated equivalently.

### 4.1. Deterministic Approach

Structures with some flutter requirements (constraints) can be optimized using computationally different strategies determined by the introduction of various forms of objective functions. Optimization (maximization) of flutter speed (bound–see Figure 1) is a central concept in the design of plated/shell structures. However, the formulation of the problem can be carried in different way given below:The direct formulation of the problem (Muc, Flis [14]):
(11)$$Max\lambda \left(s_{i}\right)$$

The implicit formulation with a bound (Song, Li, Carrera, Hagedorn [29]):

(12)$$Max\sqrt{\lambda^{2}\left(s_{i}\right)-\lambda_{e}^{2}}$$

The implicit formulation with a bound (Guo [30]):

(13)$$Min{\left[1-\frac{\lambda \left(s_{i}\right)-\lambda_{e}^{}}{\lambda_{e}^{}}\right]}^{2}$$

The maximization of weighted sum of the critical aerodynamic pressures under different probability density function of flow orientations (Li, Narita [31]):

(14)$$Max\int_{-\infty }^{\infty }\lambda \left(s_{i}=\vartheta_{\infty }\right)c\left(s_{i}=\vartheta_{\infty }\right)d\vartheta_{\infty }\;\mathrm{where}\;\int_{-\infty }^{\infty }c\left(s_{i}=\vartheta_{\infty }\right)d\vartheta_{\infty }=1$$

The maximization of natural frequencies related to the vibration modes involved in the flutter phenomenon called as aeroelastic optimization method by finite difference (AOMFD). (De Leon et al. [32], the frequency regulation problems are also discussed by Katsikadelis, Babouskos [33])

(15)$$Max\left[\omega^{2}\left(\lambda \left(s_{i}\right)\right)\right]$$

A minimum weight wing *W* subject to divergence/flutter constraints (Kameyama, Fukunaga [34])

(16)$$MinW\;\mathrm{where}\;\lambda \left(s_{i}\right)\leq \lambda_{e}$$

Maximization of the flutter critical parameter *Q_crit_*, i.e., a function of the panel’s stiffnesses, damping, and dynamic pressure of the free-stream. (Vijay, Durvasulah [35])

(17)$$MaxQ_{crit}\left(s_{i}\right)$$

The uncertainty problem—the minimization of the additional masses *w_i_* added to the wing construction and satisfying the frequency constraint (Kuttenkeuler, Ringertz [36])

(18)$$Min\left[W+\sum_{i=1}^{12}\left(w_{i}=s_{i}\right)\right]\;\mathrm{and}\;\omega_{}^{2}\left(s_{i}\right)\leq \omega_{e}^{2}$$

### 4.2. Reliability Analysis

In the aeroelastic analysis the reliability based design optimization problems constitute a special class of optimization tasks that are formulated in quite different manner that the deterministic objective functions discussed, previously including also the problem (18). The general approach to this class of problems can be found in Refences [37,38]. 

The combined use of sophisticated measurement techniques, computational simulations, and a numerical identification tool for the model parameters may result in a powerful method for material characterization of mechanical properties and damage analysis. However, in such an analysis the knowledge of a great number of various factors is required and the validity of a pure deterministic modeling is always not complete because of the scatter of experimental data. Thus, many engineering problems connected with the use of composite materials are (or may be) too complex and ill-defined to be modeled by conventional deterministic procedures. On the other hand, the use of statistical analysis is also limited to: (1) The number of experimental data (the extension of random fields of variables); (2) the definition of covariance matrices since the majority of random variables are correlated in an arbitrary manner; and (3) computational efforts in the analysis of multidimensional statistical (stochastic) problems.

In general, reliability based optimization problem ensures that failure does not occur above the allowable probability, i.e.,
(19)$$MinP\left[f_{crit}\left(s_{i}\right)\leq f_{e}\right]$$
where *P* denotes the probability, *f_crit_* is the objective function of the optimization (flutter speed, eigenfrequency, weight–see (Equations (11)–(17)), and *f_e_* means the design value of the objective function. For the sake of the simplicity of the aeroelastic analysis the problem (19) is usually rewritten as the MinMax or MaxMin problem (see Muc, Kędziora [38]) referring to the graphical representation plotted in Figure 3.

Pettit [20] pointed out that uncertainty (imprecision) should be taken into account in the case of too conservative, inefficient designs or due to adoption of new technologies and techniques. Modelling of aeroelastic characteristics with uncertain elastic moduli and Poisson’s ratio was introduced by Murugan et al. [39,40]. Liaw [41], Manan, Cooper [42], Scarth et al [43] estimated the flutter speed of a composite plate with uncertainty in the ply orientations, thickness, elastic moduli, and material and air densities. Pettit, Beran [44] analysed oscillations of an aerofoil with uncertain pitching stiffness and angle of attack. Scarth, Cooper [45] investigated the influence of material variability on stability margins in flutter characteristics. Various linear and nonlinear aeroelastic problems were formulated and solved with the aid of the reliability based design optimization methods in Refs [46,47,48,49,50,51,52,53] and even the results were compared with the results obtained with the use of deterministic optimization methods Nikbay et al. [46,52,53]. The criteria considered can include probabilistic/fuzzy constraints with both structural and aerodynamic uncertainties.

## 5. Physical (Material) Design Variables

### 5.1. Composite Materials

Considering composite multilayered fibrous composites it is possible to investigate three different categories of 2D composite constructions:Plain fibers covering the whole structure—Figure 4a;Plain fibers having the identical direction in the part of the structure—Figure 4b;Curvilinear fibers—Figure 4c.

For the planar 2D individual layers of the laminate the fiber orientation θ(*x*,*y*) is:
Constant as (x,y)∈ ΩConstant as $\left(x,y\right)\;\in \;\Omega_{i},\;U_{i}\Omega_{i}=\Omega ,\;i>1$Variable as (x,y)∈ Ω
where Ω denotes the 2D space occupied by plated/shell structures and (*x*,*y*) are the coordinates defined in the 2D curvilinear space corresponding to the plate/shell mid-surface. For fibrous composites the 2D stiffness matrices [A] (extensional), [B] (flexural–extensional coupling), and [D] (flexural) are defined in the classical way:(20)$$[A]\;=\;\int_{-t/2}^{t/2}[Q]dz,\;[B]\;=\int_{-t/2}^{t/2}[Q]zdz,\;[D]=\int_{-t/2}^{t/2}[Q]z^{2}dz$$
[Q] is the reduced stiffness constants of the materials determined from the transformation relationships for each of the *l*-th layer, separately:(21)$$\begin{array}{l} {}[Q]={\left[T^{\left(l\right)}\right]}^{Tr}[C]\;\left[T^{\left(l\right)}\right],[C]=\left[\begin{array}{ccc} E_{1}/\left(1-\nu_{12}\nu_{21}\right) & \nu_{21}E_{1}/\left(1-\nu_{12}\nu_{21}\right) & 0\\ \nu_{21}E_{1}/\left(1-\nu_{12}\nu_{21}\right) & E_{2}/\left(1-\nu_{12}\nu_{21}\right) & 0\\ 0 & 0 & G_{12} \end{array}\right],\\ \left[T^{\left(l\right)}\right]=\left[\begin{array}{ccc} c^{2} & s^{2} & sc\\ s^{2} & c^{2} & -sc\\ -2sc & 2sc & c^{2}-s^{2} \end{array}\right],\;s=\sin\left(\theta^{\left(l\right)}\right),\;c=\cos\left(\theta^{\left(l\right)}\right) \end{array}$$

For the curvilinear fiber format (Figure 4c) the number of design variables is equal to the number of parameters defining the curve *θ*(*x*,*y*)—see Muc, Ulatowska [54,55]. For angle-ply laminates (±*θ*) one design variable determines the allowable orientations in the case plotted in Figure 4a and the number of domains Ω*_i_* in the case shown in Figure 4b. The number of design variables increases for arbitrary orientations in each of the plies constituting laminate. For the Love-Kirchhoff hypothesis the laminate configuration is represented by 12 lamination parameters in the domain Ω (or each of the domains Ω*_i_*). It grows further for transverse shear deformation theories—see Muc [56]. 

It is possible to reduce the total number of design variables by introducing the pairs of discrete fiber orientations {$0_{2}^{0},\pm 45^{0},90_{2}^{0}$}—each of the plies has an identical thickness t/N. For symmetrically laminated plates/shells all terms in the stiffness matrix [B] are equal to zero and the plate/shell stiffnesses are characterized by twelve nonzero parameters, i.e., A_11_, A_12_, A_22_, A_66_, A_16_, A_26_, and D_11_, D_12_, D_22_, D_66_, D_16_, D_26_—see Muc [57]. However, they are not independent and can be expressed by four natural values: N_0_—the number of pair of plies oriented at 0^0^, *N*_90_—the number of pair of plies oriented at 90^0^ and:(22)$$\begin{array}{c} N_{0}^{D}=\sum_{l=1}^{N/4}\left[3l(l-1)+1\right]{\;\mathrm{for}\;\mathrm{plies}\;\mathrm{oriented}\;\mathrm{at}\;0}^{0}\\ N_{90}^{D}=\sum_{l=1}^{N/4}\left[3l(l-1)+1\right]{\;\mathrm{for}\;\mathrm{plies}\;\mathrm{oriented}\;\mathrm{at}\;90}^{0} \end{array}$$

In addition, knowing the values $N_{0}^{D}$, $N_{90}^{D}$ it is possible to derive the corresponding numbers of plies *N*_0_, *N*_90_ although the mapping is not unique—Muc [58]. Therefore, the space (plane) $N_{0}^{D}$, $N_{90}^{D}$ seems to be the most convenient representation of the optimization results—Figure 5. To compare discrete with angle-ply (continuous) fiber orientations it is better to introduce the following definition:(23)$$x_{D}=\frac{4}{t^{3}}\sum_{l=1}^{N}\left(z_{{}_{l}}^{3}-z_{{}_{l-1}}^{3}\right)\cos(2\theta^{\left(l\right)}),\;y_{D}=\frac{4}{t^{3}}\sum_{l=1}^{N}\left(z_{{}_{l}}^{3}-z_{{}_{l-1}}^{3}\right)\cos^{2}(2\theta^{\left(l\right)})$$

A lot of papers studied the influence of the fiber orientations, stacking sequences, and fiber mechanical properties on critical aerodynamic speed [21,59,60,61,62,63,64,65,66,67,68]. Hertz et al. [69] called the above process as aeroelastic tailoring, i.e., the design that makes use of the directional properties of fibrous composite materials in wing skins and orients these materials in optimum directions—see also References [70,71,72,73,74,75,76,77,78,79,80,81,82,83,84,85,86,87,88]. In different papers the dominant role of the bending stiffness D_11_ is observed. Vijay, Durvasulah [35] introduced the critical parameter (17) as the value divided by the D_11_ value. The similar effects were noticed by Muc, Flis [14], Rikards, Teters [15]—see Figure 6. It is interesting to note that the decrease of the aerodynamic critical pressure with the growth of the fiber orientations (the angle-ply configuration is considered) is almost insensitive to the form of boundary condition and the orthotropy ratio E_L_/E_T_.

Steering and control fiber orientations in finite elements (Figure 4b) combined with topology optimization is discussed by Muc et al. [89,90] Figure 7 shows the example of possible solutions for three discrete fiber orientations considered (i.e., {$0_{2}^{0},\pm 45^{0},90_{2}^{0}$}).

It has been noticed that the flutter maximal speed can be increased or decreased depending on the variable fiber spacing—Shih-Yao Kuo [91]. Stodieck et al. [92,93] studied the aeroelastic behavior of rectangular composite wing. Tow-steered composites were used to tailor the aeroelastic behavior, and they showed a good performance over traditional unidirectional composite laminates. The identical conclusion to the above was formulated later in References [92,93,94,95,96,97,98,99,100,101]. Stodieck et al. [94] assessed the potential wing weight savings of a full-size aeroelastically tailored wing. It turned out that optimized tow-steered laminates achieve much better mass reductions than optimized straight fiber composites. Haddadpour and Zamani [95] investigated the aeroelastic design of composite wings with curvilinear fiber which were modelled as thin-walled beams. The wing was optimized with a linear spanwise variation of the fiber orientation to maximize the aeroelastic instability speed. It was shown that much improved aeroelastic stability was achieved by the optimal variable stiffness wings compared with the conventional, constant-stiffness ones. Stanford et al. [96] studied the aeroelastic tailoring of a cantilevered flat plate in low-speed flow, locating the Pareto front between static aeroelastic stresses and dynamic flutter boundaries using a genetic algorithm. Guimaraes et al. [97] investigated the flutter behavior of tow-steered composite panels using the Ritz method combined with supersonic aerodynamic piston theory. The flutter stability boundaries for constant stiffness laminates and variable stiffness laminates were compared. Akhavan and Ribeiro [98] studied the aeroelastic instability of variable stiffness composite laminates in supersonic airflow. A third-order shear deformation theory and linear piston theory were used for structural and aerodynamic modelling, respectively. The p-version finite element method was adopted to discretize the aeroelastic model. The effects of boundary conditions, fiber angles and airflow direction on the flutter and divergence occurrence were investigated. Khalafi and Fazilati [99] developed an enhanced isogeometric finite element method to investigate the free vibration and the linear flutter characteristics of variable stiffness square and skew laminated plates. Their results were compared with those in the literature to verify the accuracy and effectiveness. Ouyang and Liu [100] researched the nonlinear flutter behavior of tow-steered composite laminates in high supersonic flow using finite element method, and investigated the effects of boundary conditions and fiber orientation on the nonlinear flutter behavior.

### 5.2. Functionally Graded Materials and Nanocomposite Structures—Thermal Protection

Since 2000, numerous investigations were devoted to the study on flutter analysis of structures made of functionally graded materials (FGMs) [32], [101,102,103,104,105,106,107,108,109,110,111,112,113,114,115,116,117,118,119,120,121,122,123,124,125] and of nanocomposites (NC) [126,127,128,129,130,131,132,133,134,135,136,137,138,139,140,141,142,143,144,145,146]. In this area the parametric investigations (not always optimal search for solutions) dealt with the analysis of flutter characteristics of structures especially in view of thermal protection. 

The stiffness and mass distribution of the wing or the blade structure both have effects on the aeroelastic properties. Functionally graded materials and nanocomposites have continuously varying properties by spatially varying the distribution of two (or more) materials—see the relations (9), (10). Various gradient strategies can be applied at the different directions of the wing/blade—Figure 8. FGMs/NCs enable changes in structural stiffness, thermal and mass distribution without necessarily requiring a geometric change in the structural geometry. 

The effective material properties are usually obtained by a linear rule of mixture, assuming the prescribed volume (density) fraction of the FG/NC material (a grading function usually in the form of the power function). 

Three types of FGM/NC constructions can be analyzed—see Figure 9:1.Ceramic/metal (FGM) structures with ceramic (C) and metal (M) isotropic properties and the prescribed form of a grading function having an unknown power law coefficient n.
(24)$$E_{FGM}=E_{m}\left(1-V_{C}\right)+E_{C}V_{C},V_{C}={\left(\frac{z}{t}+\frac{1}{2}\right)}^{n},0\leq n<\infty ,-\frac{t}{2}\leq z\leq \frac{t}{2}$$2.Sandwich structures made of a FGM core and laminated faces; the core Young’s modulus can be determined from the following relation:
(25)$$E_{FGM}=E_{s}\left[{\left(1-\frac{\rho_{FGM}^{}}{\rho_{s}^{}}\right)}^{2}\frac{\rho_{FGM}^{2}}{\rho_{s}^{2}}+\frac{\rho_{FGM}^{2}}{\rho_{s}^{2}}\right]$$
where *E_FGM_* and *ρ_FGM_* (grading function) are the Young’s modulus and mass density of the core, respectively; *E_s_* and *ρ_s_* are the Young’s modulus and mass density of the solid material.3.Carbon nanotubes (CNT) embedded in the matrix–orthotropic properties of CNTs (four material constants), Young’s modulus of the matrix, and the volume fraction and distribution of CNTs (Ref [118]).

For material 1 the total number of design variables is reduced to three (two Young’s moduli and the power law coefficient), whereas for material 2 with isotropic faces also to the identical three design variables as previously. The material 3 has seven independent design variables. For material 2 with laminated faces the number of design variables increases and it is equal to the number of variables discussed in Section 5.1 plus two characterizing the FGM core. Let us note that for three design variables the optimization studies can be replaced by the parametric analysis describing the influence of all parameters. Similarly the influence of material properties and CNTs distributions on flutter characteristics are investigated by the parametric analysis only.

The majority of works is devoted to the investigations of flutter characteristics and the aerodynamic performance of rectangular panels made of the material 1 (ceramic/metal)—References [101,102,103,104,105,106,107,108,110,114,116,119,121,124]. In two papers [111,119] the similar analysis as mentioned in the previous sequence was carried for beam structures and in one paper [123] for a pipe conveying fluid. The studies were mainly focused on the problems of the thermal protection and were limited to the parametric considerations.

Sandwich structures with FGM core (material 2) were applied in the analysis of flutter problems of beams [112], panels [110], doubly curved shells [113,117], and truncated conical shells [115].

In the numerous references shown in this section in two papers only [109,110] the authors employed optimization algorithms to investigate the flutter behavior in detail. Peng; Xiaoping [109] analyzed the rectangular sandwich structures having laminated facesheets and FGM core with three types of linearly grading strategies. The authors conducted the optimal design of facesheets using the lamination parameters. Dunning et al. [110] used the similar grading strategy as in the previous work to optimize the aeroelastic performance of an aluminum alloy and AlSiC cantilever plate. The authors implemented genetic algorithms and the Pareto method. 

The optimal choice of the grading strategy can be solved properly with the use of the topology optimization strategies. The most general approach should be based on the topology optimization of various material distributions inside the structure. For plates made of porous material De Leon et al. [32] analyzed the variable material distribution assuming the existence of two possible materials: one with the defined mechanical properties and the second (pores) with the zeroth properties.

Recently a large volume of literature has also studied the supersonic flutter behaviors of nanocomposite structures. In general, the analysis demonstrated the necessity of the use of higher order transverse shear deformation theories in order to evaluate correctly the vibrational modes—References [126,127,128,129,130]. The flutter speed of CNT-reinforced composite plated/shell structures does not always increase with an increase in CNT volume fraction.

The third material discussed in this Section cannot be classified as the classical FG material; however, it is mainly connected with the application of nanocomposites to thermal protection problems which belongs directly to the area of possible application of functionally graded materials. These studies are presented in References [118,120,125].

Asadi et al. [131,132,133,134,135,136,137] investigated the aeroelastic flutter behaviors of carbon nanotube (CNT)-reinforced composite beams, flat panels, cylindrical shells, conical shells, and truncated conical curved panels in supersonic airflow. The flutter behavior of nanocomposite (CNT) cylindrical shells was studied by Zhang et al. [138,139,140,141,142]. Various physical problems connected with nanocomposites, flutter, and fluid behavior were discussed in References [143,144,145,146,147].

### 5.3. Piezoelectric (PZT) Patches–Active and Passive Flutter Control

A broad discussion of optimization problems (vibration and buckling) and methods encountered in the analysis of laminated structures reinforced by PZT patches was presented in References [17,148,149]. Usually, the optimization technique was applied to find the best geometry of the location of PZT actuators/sensors on the top/bottom surfaces of structures. The basic structure to be investigated is shown in Figure 10 where a panel, either flat or cylindrical, has two equal piezoelectric layers attached to the top and bottom surfaces. This voltage makes the laminate structure deform. Both the shape and the location of PZT patches are treated as design variables.

Usually, in optimal design of structures with S/As two forms of the response function are considered. They are directly connected with the active control of structural deformations. The first type of the response function *R* can be identified with the magnitude of displacements (total or components)—see Figure 2. Such an optimization problem can be formulated both for static and dynamic problems (active damping). In the second case the natural frequency can be treated as the response function. Now, the application of active control is becoming increasingly important especially for flutter characteristics (Figure 1). Piezoelectric materials have been applied in structural vibration control to take advantage of their fast response, their flexibility to be used as S/As in a large variety of applications, and the fact that they provide broadband frequency responses. Smart structures, which contain main structures and distributed piezoelectric S/As, can simultaneously detect certain vibration modes and generate control forces to reduce the vibration of the structures.

Various problems referring to the flutter suppression by PZT patches (applying passive or active control to eliminate instability) were studied in References [150,151,152,153,154,155,156,157,158,159,160,161,162,163,164,165,166,167,168,169,170,171,172,173,174,175,176]. Even up to a 49% flutter velocity enhancement has been achieved by using piezoelectric actuation.

The use of piezoelectric control of dynamic instability seems to be a powerful tool in changing the flutter characteristics. However, we do not intend herein to discuss the achievements of different approaches but to indicate only this problem and to demonstrate the variety of existing works. In our opinion, it is necessary to emphasize only that the location of the PZT patches is commonly selected with the use of optimization algorithms, e.g., in the form of genetic algorithms—see Song et al. [29].

### 5.4. Sandwich Structures

Sandwich constructions usually consists of two faces which are kept separated by a core. Because of the increasing number of new materials the different materials may be utilized as the facings and the core. Some of them used in the aircraft constructions and optimized with respect to the divergence or flutter behavior are given below: lattice sandwich panel [177], blade stiffened panel [178], rotated sandwich plates [179,180,181,182,183,184,185,186], PVC foam laminated wing [187,188,189,190], honeycomb with MR pockets [191], sandwich panels with CNT facesheets [192,193], FGM foam—Section 5.2.

The additional design variables introduced by the structure of sandwich constructions are directly connected with the form of cores. At the end of the present section we intend to summarize the number and the form of design variables characterizing the different materials used in the construction of structures that may fail by the loss of dynamic instability (flutter). They are presented in Table 2.

For material design variables the optimization algorithms are commonly employed in the derivation of optimal stacking sequences of laminates and in searching for the optimal positions of piezoelectric actuators. The optimization algorithms can be also used in defining of grading functions. In this case the topology optimization methods should be adopted to this class of problems.

## 6. Geometric Design Variables

Wings are the most critical elements in the structural systems of aircrafts. In general, problems encountered in this area are connected with the geometric representation of two parameters/quantities: (1) the wing section parameters and (2) the wing profile parameters. To reduce the complexity and the cost of optimization process it is necessary to introduce an appropriate parameterization. In References [194,195] it is demonstrated that the distributions of design variables mentioned above can be written as 2D curves representing not only the shape of the edge of the structure or its mid-surface, but also dimensional optimization (e.g., thickness) or topology optimization (distributions: density, volume fraction of the constituents or material with different Young’s modulus). The general method of constructing curves on the *x-y* plane is demonstrated in detail using cubic spline functions of various types and the theoretical introduction is supplemented by the numerical solutions of different problems. It is worth to mention that Skinner, Zare-Behtash [196] discussed state-of-the-art in aerodynamic shape optimization methods in a different manner that are presented herein. 

### 6.1. Cross-Section Parameters–Variable (Stepped) Thickness

The cross-section of a wing is characterized by many parameters describing not only the size of the profile (the skin thickness, spar or rib sectional area) but also the shape of the profile (rectangular, parabolic), rib/spar locations, and the total number of spars/ribs—see References [197,198,199,200,201]. The fundamental aim of the analysis is to design the thickness distribution satisfying the criterion (16) (the minimum weight of a structure). Several studies were performed in this area. The solutions of such optimization problems were demonstrated in References [34,202,203,204,205,206,207,208]. The similar analysis was carried out for the tiltrotor composite wings (complicated shapes of the blade)—References [209,210,211,212,213,214,215,216,217,218].

### 6.2. Form of the Structure

Aeroelastic characteristics of subsonic and supersonic structures are analyzed for different forms of 2D mid-surfaces and types of loading and boundary conditions. Now, we intend to distinguish the fundamental forms of structures considered in the literature and to present the supplementary references devoted to structural analysis, but not always to the optimization problems in the discussed area. 

Usually, in view of the aeroelastic behavior the following structures are investigated:Beams—References [219,220,221,222,223,224,225,226,227,228,229,230,231,232,233,234,235,236],Flat plates (triangular, rectangular, trapezoidal)—References [237,238,239,240,241,242,243,244,245],Panels: cylindrical [246,247,248,249,250,251,252,253,254,255], spherical [256,257], truncated conical [258,259,260,261,262,263],Cylindrical shells—References [264,265,266,267,268,269,270].

In our opinion, the work presented in this section should help the readers to recognize the differences in the approach to different types of structures and maybe to adopt the approaches to their own optimization problems.

## 7. Numerical (Finite Element) Packages

The numerical (finite element) study on supersonic panel flutter has received much attention in literatures. Finite element method to panel flutter analysis by Olson [271] in 1967 and then the review of existing works was presented by Bismarck-Nasr [272,273]. Many of the researchers have built their own procedures to analyze flutter problems. The flutter analysis can be also investigated with the use of commercial FE packages as e.g., Ansys, Nastran, Abaqus, or Nisa II Family of Programs (Nisa/Areo). Recently, the package ZEUS (ZONA) [274,275] is a new proposal for aeroelastic design/analysis.

The advantages of the application of the finite element packages are obvious since they allow us:To introduce different variants of boundary conditions;To investigate arbitrary laminate configurations with no elimination of the B_ij_, A_16_, A_26_, D_16_, D_26_ terms in the stiffness matrices;To use the first order transverse shear shell/plate theories instead of the simplest Love-Kirchhoff theories.

However, they have also different disadvantages and the following problems can be emphasized as the most significant:The problems with the accuracy of solved flutter problems;The problems with the solution of the optimization problems, particularly for laminated structures where non-uniqueness of solutions is commonly encountered.

## 8. Optimization Algorithms

Various optimization algorithms can be applied in the optimization of aerothermoelastic problems. Usually, the quality of the optimized design depends upon the quality of the starting point. Thus, to enhance the quality of preliminary design it is better to apply gradient free/heuristic methods, such as for instance genetic algorithms, evolutionary algorithms, particle swarm optimization methods or simulated annealing—see Muc [22,89,276]. It is not expected to improve (reduce) the computational time needed for the search but it is necessary to obtain better than for starting points values (not necessarily global optimum) of the objective function/functional.

A lot of efforts have been put into the introduction and application of effective optimization algorithms in the design of 2-D or 3-D laminated plated and shell structures. On the other hand, because of anisotropy and inhomogeneity these structures show rather complicated states of stresses and strains such that several adequate mechanical models for composites structures have been developed in recent years—see e.g., [1,2]. However, those models may be successfully implemented into optimization problems using FE models and procedures only since analytical approach is impossible (e.g., for complicated loading and boundary conditions) or may lead to very complicated formula. Such an analysis have been conducted by the author and presented in References [3,6]. In the present paper we intend to focus our attention on the extension of the optimization analysis into the field of the uncertainty of mechanical properties of composite materials. It will be done with the use of the fuzzy set—the details of the formulation of various mechanical problems for composites in a fuzzy set environment are presented in Reference [7].

A unified, consistent approach to various optimization problems in a fuzzy environment is based on an appropriate conjunction of four elements: (1) Definition and coding of design variables describing the analyzed structures, (2) optimization algorithms, (3) a FE code and (4) a fuzzy set formulation of variability (uncertainty) of mechanical properties. The latter problem is formulated with the use of appropriate membership functions. All abovementioned elements are or may be formulated independently and this is the significant advantage of the proposed approach since using the proposed methodology it is possible to formulate and solve completely different optimization problems. The flowchart of the general optimization method with the use of the numerical FE approach is demonstrated in Figure 11. 

## 9. Concluding Remarks

A unified, consistent approach to various optimization problems subjected to flutter constraints is presented and discussed in the work. By combining optimization objectives (any of those written by the relations (11)–(19)) with the assumed forms of the design variables (materials–Section 5 and geometric–Section 6) the selected objective functions/functional can be found with the aid of the numerical code (Section 7). The allowable set of equality and inequality constraints is included during the evaluation of the objective. Then implementing optimization algorithms or varying values of design variables the fitness landscape (demonstrating variations of the objective with design variables) can be built and the global optimum can be found. 

The presented review demonstrates evidently that:1.The majority of considerations is based on the parametric investigations by observing the influence of various effects on values of objective functions; it is especially visible in problems dealing with geometric design variables;2.The optimization algorithms (evolutionary techniques) are mainly employed in three groups of problems:
a.Searching for the optimal stacking sequences in laminated structures or sandwich structures with laminated facesheets; in the paper a special attention is focused on the reduction of the total number of design variables for multilayered laminate constructions;b.Location and final shapes of piezoelectric actuators/sensors used in the active or passive control of the structural response;c.Variable thickness optimization of structures.3.The broader use of optimization methods is, in our opinion, required in the following class of problems:
a.Shape optimization of structures considered, particularly in view of their loss of dynamic stability (geometric design variables);b.Topology optimization of grading functions introduced for ceramic/metal (functionally graded materials) and/or nanocomposites reinforced with carbon nanostructures

We hope that the presented review can help the readers in choosing their own path in the optimal design of structures subjected to flutter constraints. 

## Figures and Tables

**Figure 1 materials-12-04215-f001:**
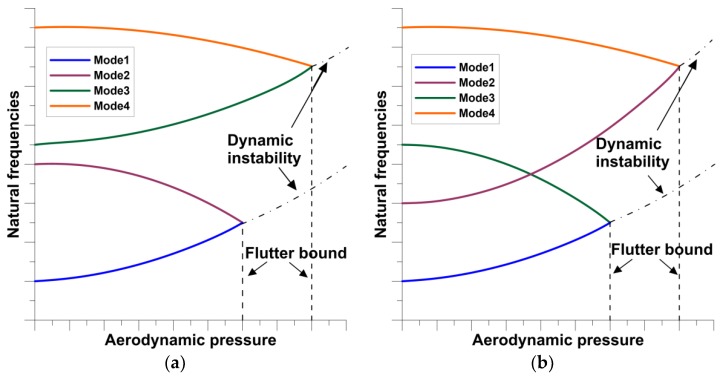
Variations of the natural frequencies with aerodynamic pressure. (**a**) Isotropic plates; (**b**) multilayered laminated plates.

**Figure 2 materials-12-04215-f002:**
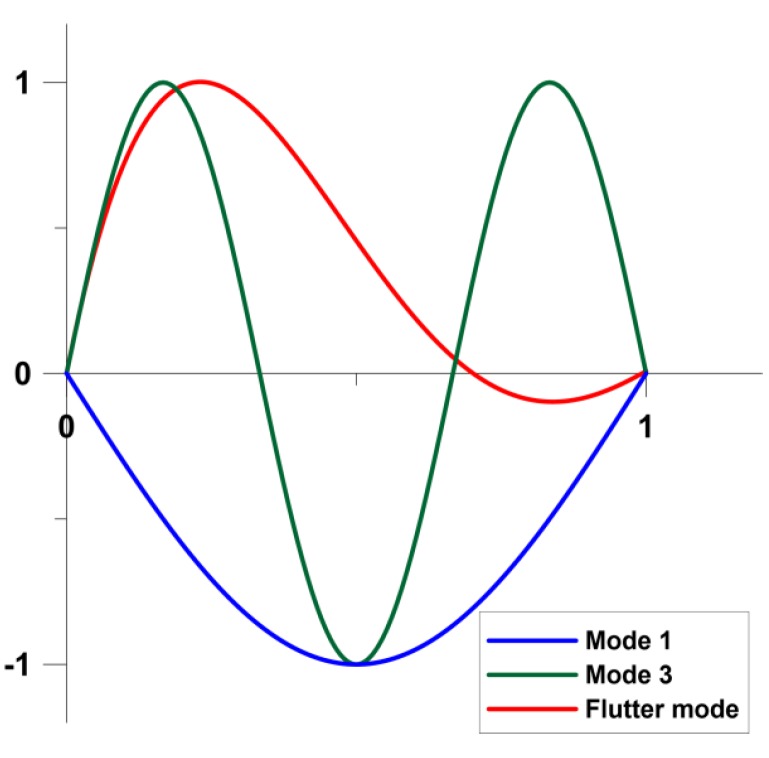
Schematical representation of the pre-flutter (free vibrations) and post-flutter modes (1D cross-section).

**Figure 3 materials-12-04215-f003:**
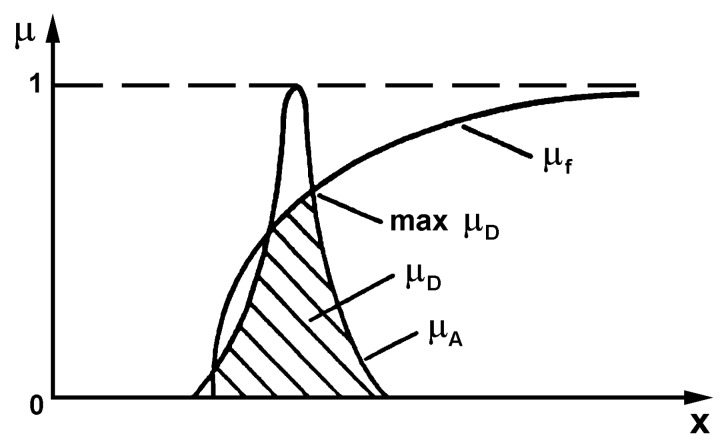
Optimal fuzzy logic solution.

**Figure 4 materials-12-04215-f004:**
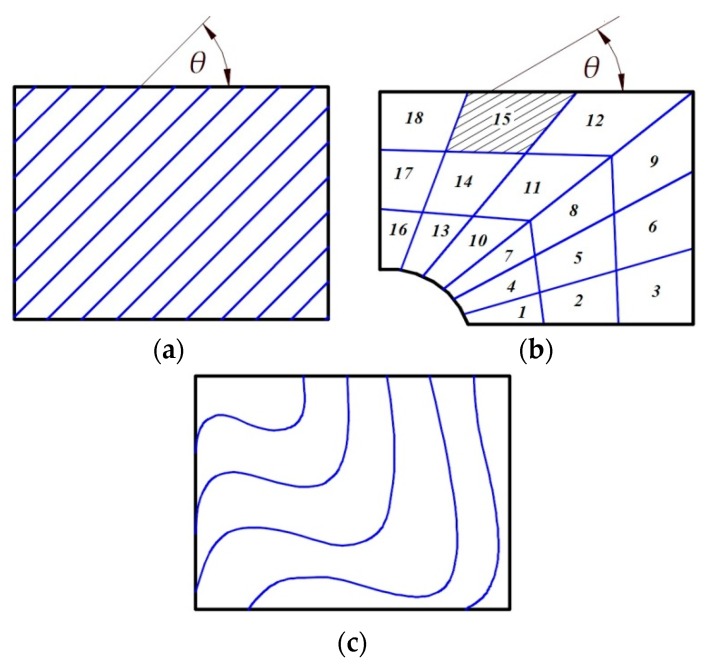
Possible constructions of the individual layers made of 2D fibers: (**a**) plain fibers, (**b**) plain fibers in the fragment of the structure, (**c**) curvilinear fibers.

**Figure 5 materials-12-04215-f005:**
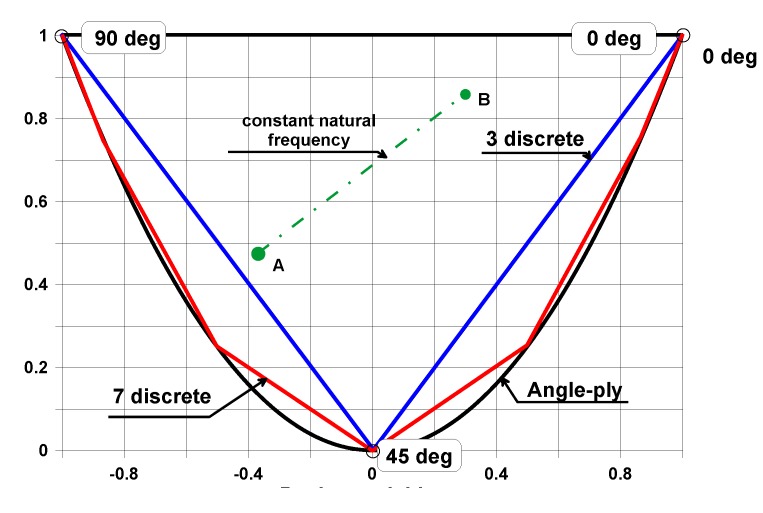
The design space for discrete and continuous fiber orientations.

**Figure 6 materials-12-04215-f006:**
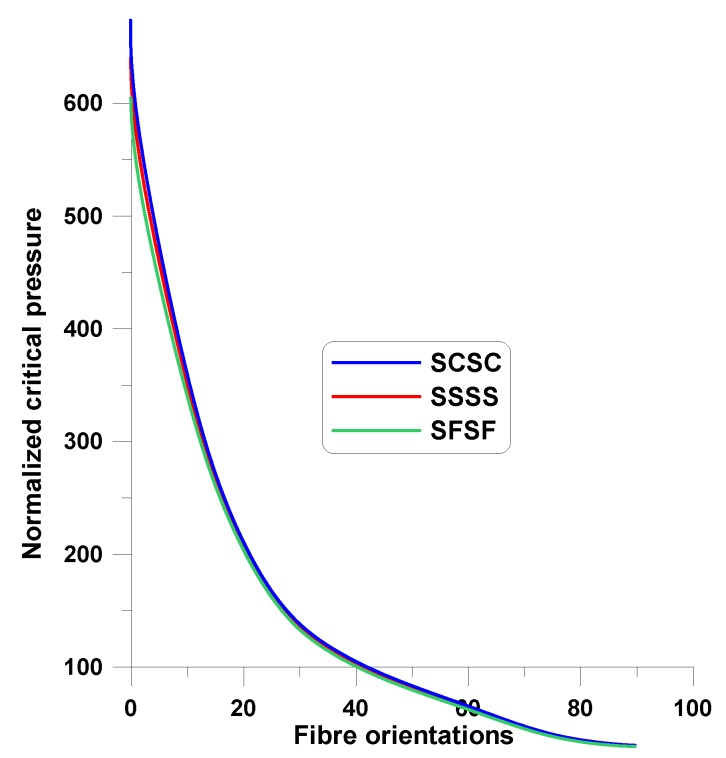
Variations of the normalized critical pressures for square angle-ply plates (L_x_/L_y_ = 1, E_L_/E_T_ = 40).

**Figure 7 materials-12-04215-f007:**
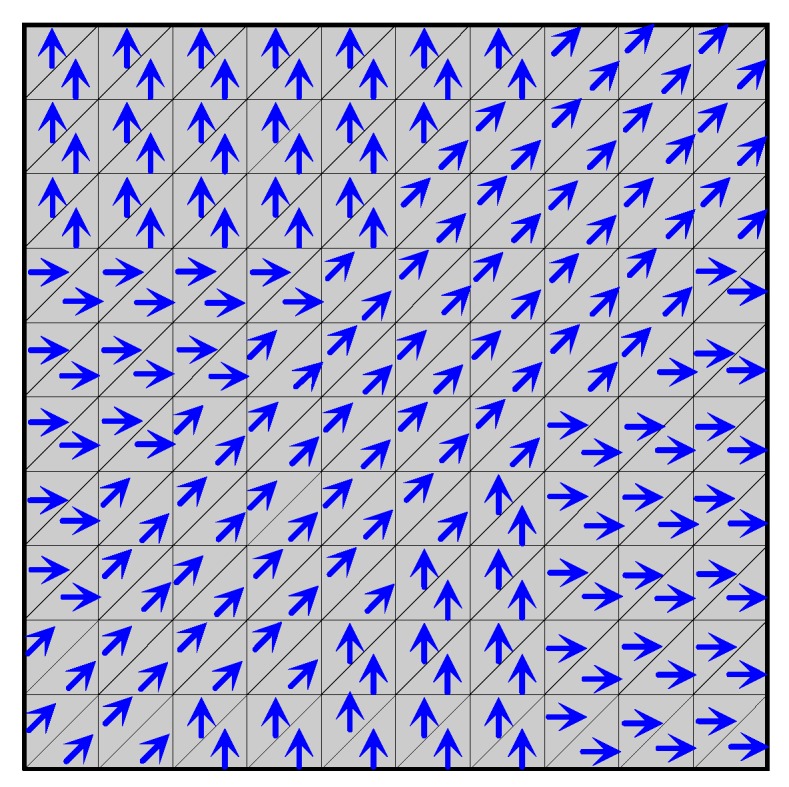
Optimal fiber orientation in the flat square plate.

**Figure 8 materials-12-04215-f008:**
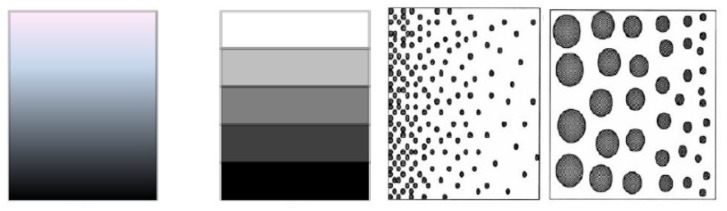
Property grading across the chord-wise/span-wise/thickness direction.

**Figure 9 materials-12-04215-f009:**
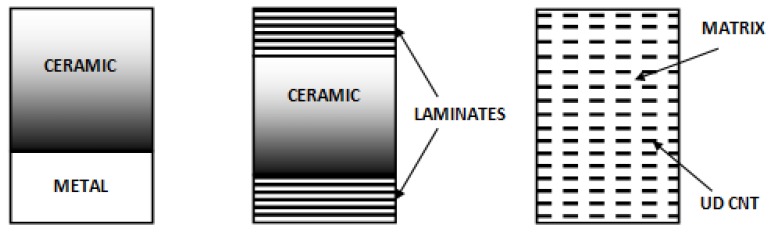
Forms of the material constructions used as the thermal protection.

**Figure 10 materials-12-04215-f010:**
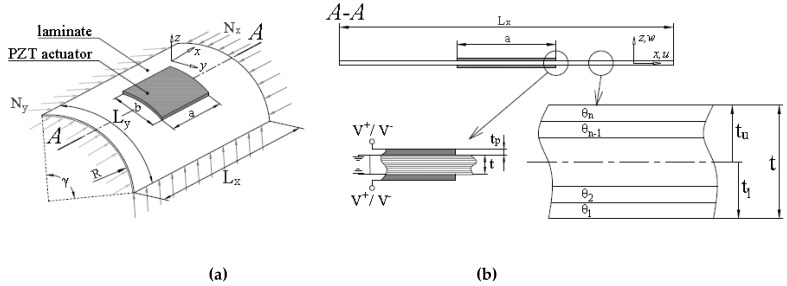
Illustration of: (**a**) The laminated panel and (**b**) the laminate cross-section.

**Figure 11 materials-12-04215-f011:**
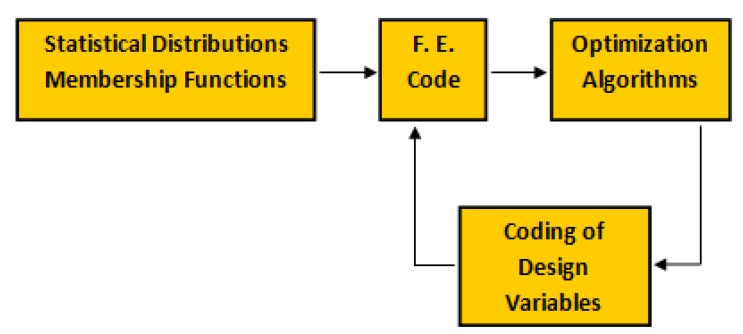
The flowchart of the general, numerical optimization scheme.

**Table 1 materials-12-04215-t001:** Classification of flutter models.

Type	Structural Theory	Aerodynamic Theory	Mach Number M
1	Linear	Linear piston theory	$\sqrt{2}\leq \;M\;\leq$5
2	Linear	Linear potential theory	1≤ M ≤5
3	Nonlinear	Linear piston theory	$\sqrt{2}\leq \;M\;\leq$5
4	Nonlinear	Linear potential theory	1≤ M ≤5
5	Nonlinear	Nonlinear piston theory	M > 5
6	Nonlinear	Navier-Stokes equations	Transonic, supersonic, hypersonic

**Table 2 materials-12-04215-t002:** Number of material design variables.

Multilayered Composite Laminates			FGM	Nanocomposites	PZT	Sandwich
Angle-Ply	Discrete 0/45/90	Curvilinear Fiber Format				
1	4	Parameters defining the characteristic curve	Mechanical properties of the constituents Parameters defining the grading function		Voltage Positions of the patches	Mechanical properties of the core

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
