# Peer review of "Optimal Design of Plated/Shell Structures under Flutter Constraints—A Literature Review"

_materials, 2019, doi:10.3390/ma12244215_

Round 1
Reviewer 1 Report
Dear Dr. Muc, Dr. Flis, and Dr. Augustyn:
Thank you for sharing your manuscript with the reviewers. The manuscript provides an extensive and impressive list of references for both optimization and flutter conditions. Unfortunately, the overall motivation and purpose of the manuscript is unclear. The link between the flutter condition and various optimization approaches is not explained fully and the many sections of the manuscript are poorly connected.
While the nature of review articles is to provide an overview of the field, such articles should highlight the key and/or latest results or methodologies that should be propagated. Your manuscript does not highlight such results or methodologies in a clear fashion. Also, the manuscript does not have a conclusion section so when the reader reaches the end of the paper, there is no summary of key points.
The overall scale of the manuscript is too broad, and the writing frequently uses bullet points to enumerate techniques without describing the value of the techniques.
Author Response
Thank you very much for your remarks. We have added additional comments and explanations.
An arbitrary optimization problem is built of three separated bricks: a) form of the objective, b) form of design variables, c) optimization algorithms. We have intended to present these elements for flutter problems. To highlight and emphasize the above we added the sentences at the end of Section 1 (l. 75-82) and Figure 11.
It is difficult to discuss the scale of the manuscript. For the comparison we added the information about other reviews – l. 71-74.
Reviewer 2 Report
The paper is a survey paper for optimal design of plated/shell structures with respect to flutter phenomena. The paper gives a large body for literature of the literature of the field, perhaps even too big which could be reduced.
What I miss in the paper is a deeper comparison of advantages and disadvantages of the many approaches available. This could give a lot of valuable to the reader as some sections feel more like just listing of the existing methods.
Also, a conclusion part is missing in the paper.
I recommend accepting the paper once these comments are taken into account in the revised version of the paper.
Author Response
Thank you very much for your remarks. We have added additional comments and explanations. However, it should be emphasized that we intend to demonstrate only the readers the present state of the art in this area. It is very difficult to discuss about the advantages and disadvantages of the works. First of all our opinions (critical or not) should be proved and secondly in our opinion the presented references deal with different problems joined only by one common word flutter.
Reviewer 3 Report
This is a interesting topic and worth being reviewed systematically. This work mainly covers the flutter-oriented design optimizations for structures of Composite and Advanced Materials, and this concentration should be revealed in the title. The first half of manuscript looks good and different literature are discussed with details. However, starting from section 5.2, it becomes a brief summary and pile of literature, which is good for an introduction section of a paper, but may be too concise for a review paper. I guess this could be due to too many sub-topics are included and the limitation of the paper length. Therefore, I recommend the authors to improve and enrich this part.
Author Response
Thank you very much for your remarks. We have added additional comments and explanations and in our opinion the paper was improved and enriched. The total number of pages increased from 26 to 31.
In our opinion all optimization problems can be discussed considering the geometrical design variables only so we do hope that it is not necessary to limit the review to composites or advanced materials in the title. We extended the chapter 5. To explain in details the broad discussion of laminated stacking sequences is necessary since it is necessary to reduce number of design variables. , e.g. from 3Nlayer/4 to four only. For other types of design variables such an operation is impossible
Round 2
Reviewer 1 Report
Dear Dr. Muc, Dr. Fils, and Dr. Augustyn:
Thank you for sending your revised manuscript. Unfortunately, the organization and purpose are still confusing. On lines 84 to 86 you wrote
The fundamental aim of the present paper is to review the existing literature on optimization of constructions subjected to flutter constraints. A special attention is focused on the proper choice of material properties since they have an important effect on flutter behavior of turbine blades or aircraft wings.
Then on line 568 you wrote
In the present paper we intend to focuse our attention on the extension of the optimization analysis into the field of the uncertainty of mechanical properties of composite materials. It will be done with the use of the fuzzy set – the details of the formulation of various mechanical problems for composites in a fuzzy set environment are presented in Ref [7].
Last, on line 583 you wrote
A unified, consistent approach to various optimization problems subjected to flutter constraints is presented and discussed in the work.
These three statements indicate three different purposes of the manuscript. In the current organization, an extension of the optimization analysis to uncertainty nor a unified approach to optimization problems is articulated. If the purpose is to discuss a new approach to optimization problems, that must be clearly articulated at the start of the manuscript and a framework presented. The bulk of the manuscript still reads as a literature review with disparate components rather than supporting an overall argument.